# The Relationship between Gut Microbiome and Cognition in Older Australians

**DOI:** 10.3390/nu14010064

**Published:** 2021-12-24

**Authors:** Mrudhula Komanduri, Karen Savage, Ana Lea, Grace McPhee, Karen Nolidin, Saurenne Deleuil, Con Stough, Shakuntla Gondalia

**Affiliations:** 1Centre for Human Psychopharmacology, Swinburne University of Technology, Hawthorn, VIC 3122, Australia; ksavage@swin.edu.au (K.S.); analea@swin.edu.au (A.L.); grace.mcphee@gmail.com (G.M.); knolidin@swin.edu.au (K.N.); saurenne@gmail.com (S.D.); cstough@gmail.com (C.S.); shakuntla.gondalia@csiro.au (S.G.); 2Health and Biosecurity, Commonwealth Scientific and Industrial Research Organization, Adelaide, SA 5000, Australia; 3Precision Health Future Science Platform, Commonwealth Scientific and Industrial Research Organisation, Adelaide, SA 5000, Australia

**Keywords:** gut microbiome, cognition, gut–brain axis, ageing

## Abstract

Ageing is associated with changes in biological processes, including reductions in cognitive functions and gut microbiome diversity. However, not much is known about the relationship between cognition and the microbiome with increasing age. Therefore, we examined the relationship between the gut microbiome and cognition in 69 healthy participants aged 60–75 years. The gut microbiome was analysed with the 16S rRNA sequencing method. The cognitive assessment included the Cognitive Drug Research computerised assessment battery, which produced five cognitive factors corresponding to ‘Quality of Episodic Secondary Memory’, ‘Quality of Working Memory’, ‘Continuity of Attention, ‘Speed of Memory’ and ‘Power of Concentration’. Multiple linear regression showed that the bacterial family *Carnobacteriaceae* explained 9% of the variance in predicting Quality of Episodic Secondary Memory. *Alcaligenaceae* and *Clostridiaceae* explained 15% of the variance in predicting Quality of Working Memory; *Bacteroidaceae, Barnesiellaceae, Rikenellaceae* and *Gemellaceae* explained 11% of the variance in Power of Concentration. The present study provides specific evidence of a relationship between specific families of bacteria and different domains of cognition.

## 1. Introduction

There has been increased research activity addressing the role of the gut microbiome in human health, and, more recently, a relationship with psychological and brain outcomes has been established. Observational studies conducted across the lifespan evidence a reduction in microbial diversity with increased age, influenced by lifestyle changes, such as diet, immunity, reduced exercise and mobility and modifications in gut morphology and physiology [1,2,3,4,5,6,7]. The microbiome in older people has a higher abundance of pathogens and reduced beneficial bacteria levels, affecting the microbial diversity. Studies on germ-free mouse models have demonstrated a relationship between gut microbes and the brain including increased blood–brain barrier permeability [8]. Further, studies of germ-free mice have reported increased anxiety and reduced neurotrophic factors such as brain-derived neurotrophic factor (BDNF) [9,10]. This alteration in the BDNF levels can be an important factor in cognitive decline. In a study of a mouse model of Alzheimer’s, when animals were treated with antibiotics (such as gentamicin, vancomycin, metronidazole, neomycin, ampicillin, kanamycin, collistin and cefaperazone), the amyloid depositions were reduced along with a reduction in species richness and increasing abundances of *Lachnospiraceae* [11] in the gut. These studies suggest a possible role of the gut microbiome and the brain structure and functions.

Research on rodent neurodegenerative disorder models such as Alzheimer’s disease and probiotic intervention studies have provided initial evidence for a gut microbiome–cognition relationship. It has been hypothesised that these effects are due to age-related changes in the microbiome [12,13,14,15]. In addition, human intervention trials of probiotics have demonstrated improvements in cognitive function, although the findings are not consistent [16]. For example, 12-week administration of fermented milk containing *Lactobacillus helveticus* IDCC3801 in older adults (60–75 years) improved cognitive function in the domains of attention and memory, compared with placebo [17]. In clinical groups of Alzheimer’s disease, consumption of a probiotic mixture (*Bifidobacterium bifidum, Lactobacillus casie, Lactobacillus fermentum* and *Lactobacillus acidophilus)* for 12 weeks was shown to improve Mini-Mental State Examination (MMSE) test scores compared to the control group [18]. However, a probiotic drink containing *Lactobacillus casei* Shirota mixed with water, sugar and skimmed milk powder was associated with a decline in episodic memory tasks and long-term memory in healthy adults (48–79 years) when compared with placebo [16].

Metabolites produced by the gut microbiome such as short-chain fatty acids (SCFAs) have a very critical role in various host functions including healthy gastrointestinal functions as well as neuroimmune function [19,20,21,22]. The SCFAs can cross the blood–brain barrier and reach the brain [23]. Decreased SCFA levels were reported in neurodegenerative disorders such as Alzheimer’s compared to the non-diseased controls [24]. Similarly, Unger et al. [25] reported decreased acetate, propionate and butyrate in Parkinson’s populations. Moreover, SCFAs can influence neuronal function by regulating neurotransmitters and neurotrophic factors. Acetate has been shown to increase anorectic neuropeptide expressions and influence the expression of neurotransmitters such as glutamate, glutamine and GABA [26]. Additionally, SCFAs can also exert anti-inflammatory effects [27]. These studies suggest an influential role of SCFAs in the brain and cognition.

There is inconsistency in previous research, together with a lack of clarity on mechanisms by which the microbiome and cognition are associated, and therefore more research is urgently required. In the present study, we utilised several gut microbiome markers including microbial diversity, the *Bacteroidetes/Firmicutes* ratio, bacterial abundance and bacterial functions to evaluate the relationship between the gut microbiome and cognition in a healthy older population. To our knowledge, this is the first study to assess the relation between the gut microbiome and cognitive domains, derived from a well-validated and standardised computerised assessment system using the Cognitive Drug Research battery (CDR).

## 2. Materials and Methods

### 2.1. Study Participants

The present study was conducted as part of an ongoing research project at the Centre for Human Psychopharmacology, Swinburne University of Technology, the Australian Research Council Longevity Intervention (ARCLI) [28,29]. The study was approved by Swinburne University Human Research Ethics Committee (SUHREC) and registered at the Australian New Zealand Clinical Trials Registry (ANZCTR12611000487910). The study recruited healthy volunteers aged 60–75 years. Participants were excluded from enrollment in the study if they had a recent history (past five years) of chronic or severe illness lasting longer than six weeks, or a psychiatric, neurological, endocrine, gastrointestinal, medically managed cardiovascular or food metabolism disorder.

Participants were non-smokers, not taking psychoactive or cognitive-enhancing medication or supplements and drank alcohol within national guideline limits (less than 14 standard drinks per week for women and 28 standard drinks per week for men). Participants were also excluded based on global cognitive impairment (defined as a score less than 24 on the Mini-Mental State Examination) [30] and depressive symptoms (defined as a score greater than 19 on the Geriatric Depression Scale) [31]. In addition, the General Health Questionnaire (GHQ-12) was administered to assess psychological distress. The scoring method was Likert, 0–3. According to Goldberg et al. [32], a score >11 indicates high levels of distress, and scores below ten are denoted as low distress. Participants were excluded from the microbiome analysis if they had taken any antibiotics and/or probiotics in the last three months prior to sample collection. Although ARCLI was an interventional study, the present sub-study used data taken from the baseline visit. The present study had a cohort size of 69, with 34 male and 35 female participants.

### 2.2. Faecal Sample Collection

Participants were provided with a faecal sample collection kit by the research team consisting of a sample collection vessel (Sarstedt, Australia), EasySampler stool collector (GP Medical Devices, Denmark), pair of gloves and a pre-frozen ice pack during the training visit (visit one). Participants were asked to collect the samples in the collection vessel (provided) preferably a day before their scheduled visit to the Centre for Human Psychopharmacology. The collection vessels with samples were then placed in a sealable plastic bag (provided) along with a pre-frozen ice pack (provided) and stored in a freezer before returning to the Centre for Human Psychopharmacology. All the faecal samples collected from the participants were stored at −80 °C until further analysis.

### 2.3. 16S rRNA Sequencing and Data Processing

The DNA was extracted from the faecal samples using the QIAGEN stool mini kit (QIAGEN Pty Ltd., Chadstone Centre, VIC, Australia), and 16S rRNA sequencing was performed by the Australian Genome Research Facility (AGRF, https://www.agrf.org.au (accessed on 18 November 2021)). Prior to sequencing, the 16S rRNA gene was amplified along with an adapter. The amplified gene was then sequenced using the target primers (27F-519R) to generate paired-end sequences of 300 bp length. The paired-end sequences were assembled by aligning the forward and reverse sequences using PEAR (version 0.9.5) [33]. After trimming the primer region, the sequences were clustered, and the number of reads was counted using Quantitative Insights into Microbial Ecology (QIIME 1.8) [34], USEARCH (version 8.0.1623) [35,36] and UPARSE software. The reads were then mapped back to OTUs (operational taxonomic units) with a minimum identity of 97%. The taxonomy was assigned to OTUs using the Greengenes database (version 13_8, August 2013) [37]. Quality filtering and construction of full-length duplicate sequences of trimmed sequences were conducted with USEARCH tools. Singleton or unique reads were removed before assigning taxonomy.

### 2.4. Functional Analysis

Pathway abundances of the gut microbiome were predicted using Phylogenetic Investigation of Communities by Reconstruction of Unobserved States (PICRUST2, version 2.3.0) [38]. The denoised and demultiplexed sequences were used to assign functional predictions based on the enzyme commission number (EC number), Kyoto Encyclopedia of Genes and Genomes (KEGG Orthology, KO) and MetaCyc metabolic pathway database. Further, the function data were grouped into gut–brain modules using the package Omixer-rpmR in R [39]. The Omixer-rpmR workflow maps gene abundances in a predefined gut-specific module database and quantifies the human gut metabolic pathway module for each sample. Each gut-specific module consists of related enzymatic functions representing a cellular process with specific input and output metabolites.

### 2.5. Cognition

Cognitive function was measured using the Cognitive Drug Research computerised assessment system (CDR) test battery (University of Reading, Berkshire, UK). The CDR battery measures five validated cognitive factor scores corresponding to ‘Quality of Episodic Secondary Memory’, ‘Quality of Working Memory’, ‘Continuity of Attention’, ‘Speed of Memory’ and ‘Power of Concentration’. The scores of the individual tasks were summed into factors based on Wesnes et al. (2000). The CDR system has been used in thousands of drug trials and has excellent age-related normative data [40]. The model used to calculate the cognitive factors is listed in Appendix A.

### 2.6. Statistical Analysis

Statistical analyses were carried out in SPSS version 26 and R version 3.6.3 using the packages phyloseq [41], ggplot2 [42], microbiome [43], dplyr [44] and vegan [45]. The mean and SD for measures of participant characteristics were calculated using the ‘mean’ and ‘sd’ functions in R packages ‘base’ and ‘stats’, respectively. The microbiome data were represented as compositional and centred log-ratio transformed using the ‘transform’ function in the microbiome R package. The microbial families with mean abundance greater than 0 were included in the analysis. The samples with missing data were also excluded from the analysis. In the present study, the associations between the microbiome (family level) and cognition were assessed using non-parametric Spearman correlations. Significance was set at a *p* value of 0.05 for the analysis. To reduce the multiplicity and to avoid the need for correcting for multiple comparisons, Spearman correlations were performed between bacterial families and cognitive domains, and in the second step, only correlated bacterial families were chosen for the regression model with the respective cognitive domains.

The associations between taxa and cognitive variables were assessed by fitted linear regression models on centred log-ratio transformed data with the ‘glm’ R function. In addition to assessing individual taxa and cognition associations, alpha diversity indices (Observed, Shannon index, Chao1, Fisher, Simpson, Inverse Simpson, ACE) were calculated using the ‘alpha’ function in the microbiome R package. The *Bacteroidetes*/*Firmicutes* ratio was calculated using the ‘bfratio’ function in the microbiome R package. Extreme negative outliers were removed from the CDR factor Continuity of Attention.

## 3. Results

A total of 69 participants (34 male and 35 female) with an age of 65.06 ± 4.01 years were included in the present study. The sample characteristics are outlined in Table 1. The mean BMI of the cohort was 26.6 kg/m^2^. In addition, the General Health Questionnaire (GHQ-12) was administered, with a mean score of 8.7, indicating low psychological distress among the cohort. The sample had a mean score of 28.7 on the Mini-Mental State Examination (MMSE), suggesting that the cohort was cognitively healthy.

### 3.1. Gut Microbiome

The mean prevalence of each taxon was calculated (Appendix A). Twelve phyla and fifty-five bacterial families were identified in the cohort. The twelve phyla identified in the cohort were *Actinobacteria, Bacteroidetes, Elusimicrobia, Firmicutes, Fusobacteria, Lentisphaerae, Proteobacteria, Synergistetes, TM7, Tenericutes, Verrucomicrobia* and *Thermi* (Figure 1). Bacteria belonging to families *Coriobacteriaceae, Bacteroidaceae, Prevotellaceae, Rikenellaceae, Clostridiaceae, Lachnospiraceae, Ruminococcaceae, Veillonellaceae, Erysipelotrichacea* and *Alcaligenaceae* were prevalently present in the cohort.

### 3.2. Association between Microbial Diversity and Cognition

There were no significant associations between alpha diversity indices (Observed, Shannon evenness index, Chao1, Fisher, Simpson, Inverse Simpson, ACE) and cognition (Table 2). There were also no significant correlations between alpha diversity indices and demographic measures such as age, sex and BMI (Table 2). The *Bacteroidetes/Firmicutes* ratio was also calculated in order to assess correlations with the cognitive scores. There were no significant correlations between these variables and cognition (Table 2). Lastly, there was no significant correlation between the *Bacteroidetes/Firmicutes* ratio and demographic measures (Table 2).

### 3.3. Association between Microbial Family and Cognition

Relative abundances of the different bacterial families were considered for analyses. Out of the 56 families measured in the present study, 9 bacterial families showed a significant correlation with at least one cognitive domain (Table 3 and Figure 2).

There was a significant positive association between the bacterial family *Carnobacteriaceae* and QESM. QWM had a significant negative association with *Alcaligenaceae* and a positive association with *Clostridiaceae.* The bacterial families *Bacteroidaceae, Barnesiellaceae, Gemellaceae* and *Rikenellaceae* were positively correlated with PoC. Further, the bacterial families *Bacteroidaceae, Barnesiellaceae, Gemellaceae* and *Micrococcaceae* were positively associated with SoM. *Clostridiaceae* and *Rikenellaceae* were positively associated with CoA. Conversely, *Verrucomicrobia* showed negative associations with CoA (Table 3).

### 3.4. Association between the Gut Microbiome and Demographic Measures

The correlations between the gut microbiome variables and the demographic variables (age, sex and BMI) are summarised in Table 3. Age was negatively associated with *Barnesiellaceae* and *Lactobacillaceae* and positively associated with *Desulphovibrionaceae* and *Porphyromonadaceae*. Lower abundances of *Odoribacteraceae* (1.97 ± 1.79)*, Rikenellaceae* (5.43 ± 0.95) and *Tissierellaceae* (−2.19 ± 1.54) and higher abundances of *Prevotellaceae* (4.49 ± 3.17) were observed in males compared to females. Lower abundances of *Prevotellaceae* (3.02 ± 2.65) and higher abundances of *Odoribacteraceae* (3.02 ± 1.39)*, Rikenellaceae* (6.01 ± 1.10) and *Tissierellaceae* (−1.56 ± 1.50) were observed in females compared to males. There were no significant relationships between gut bacterial families and BMI.

Regression analysis revealed that the significant associations between the gut microbiome and cognition were not due to the demographic variables (Table 4).

### 3.5. The Combined Effect of the Gut Microbiome on Cognition

Multiple linear regression analysis was performed to evaluate the combined contribution of significant bacterial families identified from correlation analysis in predicting cognition (Table 5). A significant regression model was identified between Quality of Episodic Secondary Memory and the bacterial family *Carnobacteriaceae* (F(1,66) = 7.966, *p* = 0.006), with an adjusted R^2^ of 0.094, accounting for 9% of the variance. The bacterial families *Alcaligenaceae* and *Clostridiaceae* revealed significant relations with Quality of Working Memory (F(2,65) = 6.973, *p* = 0.002), with an adjusted R^2^ of 0.151, accounting for 15% of the variance. Lastly, the bacterial families *Bacteroidaceae*, *Barnesiellaceae, Gemellaceae* and *Rikenellaceae* significantly predicted Power of Concentration (F(4,63) = 3.031, *p* = 0.024), with an adjusted R^2^ of 0.108, accounting for 11% of the variance.

### 3.6. Relation between Gut Microbial Function and Cognition

The association between cognitive factors and the gut–brain-specific modules, consisting of related enzymatic functions representing a cellular process, was evaluated. We found that propionate production was negatively associated with the CDR factor CoA (r = −0.311, *p* = 0.011) (Table 6). Further, ‘Power of Concentration’ was negatively associated with tyrosine degradation (r = 0.274, *p* = 0.024) and phenylalanine degradation (r = 0.274, *p* = 0.024) (Table 6). The CDR factor QWM was negatively associated with tyrosine degradation (r = −0.246, *p* = 0.045) and phenylalanine degradation (r = −0.246, *p* = 0.045) (Table 6).

## 4. Discussion

In the present study, our primary aim was to assess the relation between the gut microbiome and cognitive domains as calculated using a validated and standardised computerised assessment system, the Cognitive Drug Research battery (CDR). The CDR system is designed to test the attention, executive function and working memory, episodic secondary memory, motor control and psychophysical thresholds [46], and the CDR system is also sensitive to age-related cognitive decline [40]. Further, we derived five factors representing five different cognitive domains including ‘Quality of Episodic Secondary Memory’, ‘Quality of Working Memory’, ‘Power of Concentration’, ‘Continuity of Attention’ and ‘Speed of Memory’. Grouping individual cognitive tests into factors minimises type I error [47]. The participants’ average BMI was 26.57 ± 4.76 kg/m^2^. The BMI range between 25.0 and 29.9 kg/m^2^ is, in general, considered overweight. However, according to the 2004 Australian Institute of Health and Welfare report, older Australians are proportionally likely to be categorically overweight [48]. In addition, the mean of the General Health Questionnaire (GHQ-12) was 8.7, indicating low psychological distress among the cohort [32]. Additionally, a mean score of 28.7 on the Mini-Mental State Examination (MMSE) was observed. A score of 27–30 indicates normal cognitive functioning, while scores of 24–26 indicate possible mild cognitive impairment, and scores below 24 indicate possible dementia [30]. This indicates the cohort of the current study was healthy.

As ageing impacts various biological and psychological processes. The composition of gut bacteria could be affected by factors such as ageing [4], BMI [49] and possibly gender [50]. However, we did not find any significant associations between alpha diversity measures, cognition and demographic variables (age, gender and BMI) in the current study. Lower alpha diversity (Chao1, Phylogenetic diversity, Observed, Species richness and Species evenness) has been previously reported to be associated with poorer cognition specifically, with longer reaction times and poor verbal fluency [51,52]. In a systematic review on the gut microbiome and ageing, higher alpha diversity was reported among the oldest participants [53]. Further, the gut microbiome in older individuals is mainly dominated by the bacterial phyla Bacteroidetes and Firmicutes. Saji et al. [54] reported a higher Firmicutes/Bacteroidetes ratio in participants with dementia (defined by the authors as populations who score MMSE ≤ 20 and clinical dementia rating ≥1) compared to the non-demented control group [54]. However, we did not find an association between the Bacteroidetes/Firmicutes ratio, cognition and demographic variables.

We identified a significant association between bacterial families belonging to *Firmicutes* and cognition in the cohort. We found that higher abundances of the bacterial family *Carnobacteriaceae* were associated with better ‘Quality of Episodic Secondary Memory’, while higher abundances of *Clostridiaceae* were associated with better ‘Continuity of Attention’. Increased levels of *Gemellaceae* were associated with increased ‘Power of Attention’ and ‘Speed of Memory’. In contrast to our findings, lower abundances of *Clostridiaceae* have been observed in Alzheimer’s and mild cognitive impairment [55], and Vogt et al. [56] reported higher abundances of *Gemellaceae* in Alzheimer’s. A possible reason for this difference could be the use of cognitive domains in the present study instead of individual cognitive tests. In support of our findings, a reduction in the abundances of *Lactobacillus* and increased abundances of *Desulphovibrionaceae* and *Porphyromonadaceae* upon ageing have been reported [57,58]. Further, increased abundances of the gut bacteria *Akkermansia* and lower abundances of *Faecalibacterium, Bacteroidaceae* and *Lachnospiraceae* were identified in the oldest old [53]. However, centenarians (≤104 years) and supercentenarians (>104 years) have reduced proportions of *Bacteroides, Roseburia* and *Faecalibacterium,* while the health-associated bacterial proportions such as *Bifidobacteria* and *Christensenella* are higher [59,60]. These results suggest the impact of ageing on the gut microbiome composition. Further, we found a significant association between sex and specific bacteria (Table 3). However, we did not find any combined effect of demographic variables such as age, sex and BMI on the gut microbiome–cognition relationship when tested using an adjusted regression model.

Further, as tested by the regression model, the bacterial family *Carnobacteriaceae* explained 9% of the variance in predicting ‘Quality of Episodic Secondary Memory’. *Alcaligenaceae* and *Clostridiaceae* explained 15% of the variance in predicting ‘Quality of Working Memory’; *Bacteroidaceae, Barnesiellaceae, Rikenellaceae* and *Gemellaceae* explained 11% of the variance in ‘Power of Concentration’. Higher proportions of *Bacteroidaceae* were positively associated with ‘Speed of Memory’, and higher proportions of *Barnesiellaceae* and *Rikenellaceae* were positively associated with ‘Power of Concentration’. The bacterial families *Bacteroidaceae, Barnesiellaceae* and *Rikenellaceae* belong to the phylum Bacteroidetes. Previously, low levels of *Bacteroidaceae* have been reported in individuals with dementia and Alzheimer’s [54,61]. *Bacteroidaceae* are butyrate producers, which helps raise the levels of the neurotransmitter brain-derived neurotrophic factor (BDNF) [62]. BDNF protects the intestinal mucosal barrier (IMB) function [63]. BDNF also plays an important role in synaptic plasticity and memory processes. Administration of the probiotic VSL #3 containing eight different strains of *Lactobacillus*, *Bifidobacterium* and *Streptococcus*, in mice, increased hippocampal BDNF in both the young and aged [64], which suggests a possible role of the gut microbiome in influencing cognition via influencing neurotransmitter production. In contrast to our findings, bacteria from the *Rikenellaceae* and *Barnesiella* families have been reported in higher abundances in a small sample of 13 participants with Parkinson’s disease who showed mild cognitive impairment [65]. Our study findings suggest positive effects of *Bacteroidetes* on cognition in ageing populations.

We also mapped functional profiles of the microbiome using PICRUST2. We found that increased propionate production was negatively associated with ‘Continuity of Attention’, while tyrosine degradation and phenylalanine degradation were shown to be negatively associated with ‘Speed of Memory’ and ‘Quality of Working Memory’. The results of our study show increased propionate production associated with decreased attention. Propionate is a significant SCFA, as excess propionate in the circulation could lead to motor impairments, brain atrophy, cognitive impairments and dementia [66,67] and increase the risk of Alzheimer’s disease [68]. Administration of propionate ameliorates motor deficits and dopaminergic neuronal loss in Parkinson’s [69]. All these studies suggest a possible role of propionate in cognition. In addition, we found that tyrosine degradation and phenylalanine degradation are negatively associated with performance on speed of processing and working memory tests. This relation between tyrosine, phenylalanine and cognition is supported by research on amyotrophic lateral sclerosis, where increased phenylalanine and tyrosine levels have been detected [70]. Given the current findings in a healthy ageing population, this suggests that increased propionate production, tyrosine degradation and phenylalanine degradation could be harmful to neuronal health. However, more research is needed to validate the results.

The strengths of the present study are the study design and method used for the cognitive assessment. We evaluated the relationship between the gut microbiome and cognition by grouping individual tests into cognitive domains. The present study is the first to analyse the relationship between the gut microbiome and well-validated and comprehensive cognitive factors instead of individual cognitive tests in healthy older participants. Grouping individual cognitive tests into cognitive factors or domains helps researchers better understand and interpret past research findings [71]. Further, grouping individual test results reduces the number of statistical comparisons and the chances of type I error [72]. Hence, the present study grouped individual cognitive tests into factors. Additionally, the current study studied the relationship between the gut microbiome and cognition in a healthy ageing population (60–75 years old). The cohort was healthy and devoid of any diseases or disorders affecting cognition or gut microbial composition. Further, the cohort had low psychological distress and was devoid of depression. In addition, the cohort did not report any memory and attention problems. Identifying the associations between gut bacterial families and cognitive domains is a critical first step, but there are limitations to the current study. Factors such as diet, physical activity and lifestyle are modifiers of the gut microbiome as well as cognition. However, the current study did not assess the role of these factors. Therefore, more in-depth studies are needed for a better understanding of the relationship between the gut microbiome and cognition in healthy ageing.

## 5. Conclusions

The present study is the first to identify a relationship between the gut microbiome and comprehensive cognitive factors in healthy older populations. We grouped individual cognitive tests into cognitive domains using the CDR factor system in order to assess the relationship between different indices of the gut microbiome and cognition using five cognitive factors representing long-term memory, working memory, attention, processing speed and memory retrieval. We identified specific relationships between individual bacterial families and the cognitive domains. We found that propionate production, tyrosine degradation and phenylalanine degradation were negatively related to cognition. These results add to the growing literature on the relationship between cognition, ageing and the microbiome and suggest that pre- and probiotic interventions based on our results may be helpful in older adults with cognitive decline. The results of the present study add to the growing credibility of the gut microbiome as a potential therapeutic target for cognitive impairments.

## Figures and Tables

**Figure 1 nutrients-14-00064-f001:**
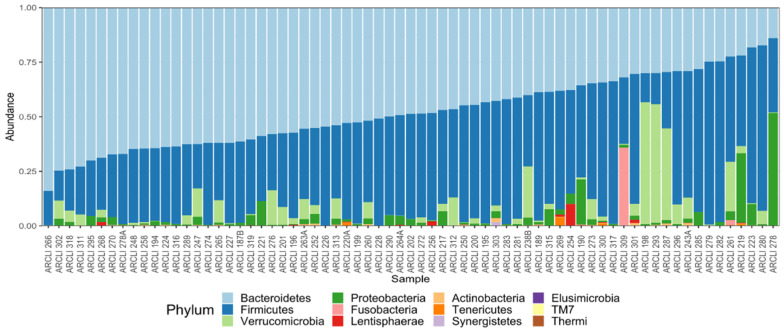
Bar plot of the abundance of different phyla in the cohort. The bar graph is presented as percent abundances of a total of twelve phyla for each participant. Graph is sorted based on *Bacteroidetes* abundance from lower to higher using the R package ‘microbiome’.

**Figure 2 nutrients-14-00064-f002:**
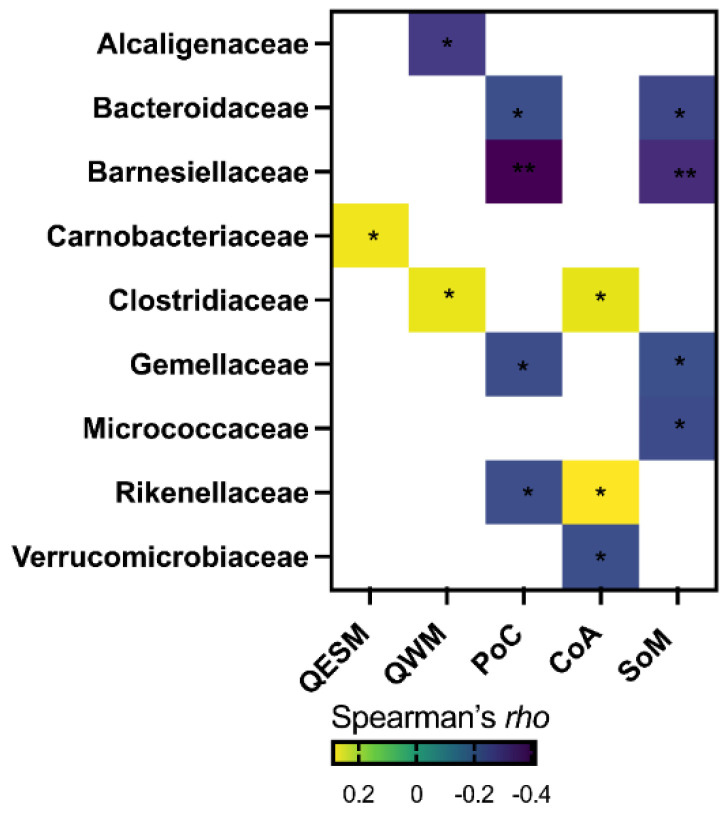
Correlations between bacterial families and cognitive domains. Correlations between the gut microbiome and cognition were conducted with the Spearman method. Only significant correlations are listed in the heatmap. A value of *p* < 0.05 was considered significant, where * *p* < 0.05 and ** *p* < 0.001.

**Table 1 nutrients-14-00064-t001:** Participant characteristics.

Characteristic	Mean	SD
Sample Size	69	
Gender	Male (34), Female (35)	
Age	65.06	4.01
BMI	26.57	4.76
MMSE	28.78	1.29
GDS	3.91	3.34
General Health Questionnaire (GHQ-12)	8.66	2.74
**Cognition**		
Word Recall Original Accuracy	70.69	16.30
Word Recall Novel Accuracy	86.96	12.10
Picture Recall Original Accuracy	92.57	8.92
Picture Recall Novel Accuracy	87.21	10.42
Immediate Word Recall Accuracy	38.53	11.91
Immediate Word Recall Error	0.32	0.63
Delayed Word Recall Accuracy	22.26	11.89
Delayed Word Recall Error	0.74	1.05
Spatial Working Memory Sensitivity Index	0.83	0.30
Numeric Working Memory Sensitivity Index	0.91	0.10
Simple Reaction Time	299.60	39.76
Digit Vigilance	441.29	49.42
Choice Reaction Time	510.88	49.32
Spatial Working Memory Reaction Time	1099.83	346.92
Numeric Working Memory Reaction Time	848.11	163.61
Word Recall Reaction Time	1006.38	194.81
Picture Recall Reaction Time	1166.98	237.06
Digit Vigilance Accuracy	96.64	6.79
Choice Reaction Time Accuracy	98.15	1.74
Digit Vigilance False Alarms	3.50	14.03
**CDR factors**		
Quality of Episodic Secondary Memory (QESM)	191.15	39.38
Quality of Working Memory (QWM)	1.77	0.26
Power of Concentration (PoC)	1251.77	99.64
Continuity of Attention (CoA)	90.88	6.76
Speed of Memory (SoM)	4115.16	718.37

BMI, body mass index; GDS, Geriatric Depression Scale; SD, standard deviation; MMSE, Mini-Mental State Examination.

**Table 2 nutrients-14-00064-t002:** Association between alpha diversity indices, demographics and cognition.

Alpha Diversity Index	Observed	Shannon	Chao1	Fisher	Simpson	Invsimpson	ACE	B/F Ratio
Age	0.157	0.066	0.085	0.084	0.103	0.103	0.081	0.006
Sex	−0.01	−0.082	0.032	−0.14	−0.01	−0.01	−0.006	−0.042
BMI	−0.097	−0.111	−0.025	−0.042	−0.136	−0.136	−0.039	0.023
QESM	−0.113	−0.011	−0.067	−0.072	0.003	0.003	−0.057	−0.035
QWM	−0.034	−0.193	−0.094	−0.042	−0.202	−0.202	−0.083	−0.171
PoC	0.011	0.079	−0.008	−0.001	0.131	0.131	−0.054	−0.144
CoA	−0.021	0.131	−0.02	−0.003	0.186	0.186	−0.075	−0.182
SoM	0.061	0.21	0.021	0.103	0.218	0.218	−0.014	−0.171

Data are presented as Spearman’s *Rho* (correlation coefficient), A value of *p* < 0.05 was considered significant. We did not find any significant relation between alpha diversity indices, *Bacteroidetes/Firmicutes* ratio and cognition, QESM, Quality of Episodic Secondary Memory; QWM, Quality of Working Memory; PoC, Power of Concentration; CoA, Continuity of Attention; SoM, Speed of Memory.

**Table 3 nutrients-14-00064-t003:** Association between bacterial family, cognition and anthropometric measures.

Bacterial Family	QESM	QWM	PoC	CoA	SoM	Age	Sex	BMI
*Alcaligenaceae*	−0.031	−0.294 *	0.103	0.119	0.198	−0.026	0.013	0.095
*Bacteroidaceae*	−0.006	0.064	−0.247 *	0.03	−0.265 *	0.015	0.173	−0.033
*Barnesiellaceae*	0.043	0.221	−0.413 **	0.109	−0.328 **	−0.283 *	0.058	0.041
*Carnobacteriaceae*	0.273 *	0.217	−0.062	−0.057	−0.238	−0.077	0.105	0.03
*Clostridiaceae*	0.229	0.265 *	−0.017	0.261 *	−0.015	0.025	0.119	−0.018
*Desulfovibrionaceae*	−0.019	−0.148	−0.098	−0.054	−0.03	0.247 *	0.007	0.1
*Gemellaceae*	−0.05	−0.029	−0.252 *	−0.111	−0.245 *	−0.051	0.127	0.156
*Lactobacillaceae*	0.121	−0.165	0.184	−0.033	0.152	−0.317 **	−0.015	−0.129
*Micrococcaceae*	0.087	0.07	−0.057	−0.093	−0.255 *	−0.102	0.016	0.054
*Odoribacteraceae*	0.073	0.123	−0.172	0.149	−0.075	−0.051	0.320 **	−0.027
*Porphyromonadaceae*	−0.026	−0.159	−0.146	−0.011	−0.183	0.240 *	−0.084	0.055
*Prevotellaceae*	−0.107	−0.168	0.163	0.032	0.126	0.13	−0.269 *	0.03
*Rikenellaceae*	0.167	0.027	−0.248 *	0.288 *	−0.075	−0.121	0.355 **	−0.174
*Tissierellaceae*	0.223	0.001	0.163	−0.132	0.014	−0.057	0.272 *	−0.096
*Verrucomicrobiaceae*	−0.051	0.008	−0.052	−0.247 *	0.139	0.08	0.025	−0.048

Data are presented as Spearman’s *Rho* (correlation coefficient), where * *p* < 0.05, ** *p* < 0.01.

**Table 4 nutrients-14-00064-t004:** Bacterial families in the prediction of cognition per domain, adjusted and unadjusted regression models.

Bacterial Family	Cognitive Domain	Unadjusted				Adjusted ^+^			
		*β*	CI (2.5, 97.5)		*p* Value	*β*	CI (2.5, 97.5)		*p* Value
*Carnobacteriaceae*	QESM	10.27	3.14	17.40	0.006	9.25	2.12	16.39	0.014
*Alcaligenaceae*	QWM	−0.08	−0.12	−0.03	0.002	−0.08	−0.13	−0.04	0.001
*Clostridiaceae*		0.05	−0.01	0.11	0.13	0.05	−0.01	0.11	0.12
*Bacteroidaceae*	PoC	−21.8	−44.57	0.98	0.07	−23.16	−46.68	0.36	0.06
*Barnesiellaceae*		−19.74	−34.78	−4.69	0.01	−21.15	−37.12	−5.13	0.01
*Gemellaceae*		−30.46	−57.30	−3.62	0.03	−31.05	−58.97	−3.14	0.03
*Rikenellaceae*		−22.85	−44.71	−0.99	0.05	−27.24	−50.47	−4.00	0.03
*Clostridiaceae*	CoA	1.32	0.06	2.57	0.0	1.23	−0.06	2.52	0.07
*Rikenellaceae*		1.26	−0.24	2.77	0.10	1.28	−0.31	2.87	0.12
*Verrucomicrobiaceae*		0.27	−0.28	0.83	0.34	0.25	−0.33	0.80	0.42
*Bacteroidaceae*	SoM	−181.87	−347.11	−16.62	0.04	−191.14	−362.41	−19.87	0.03
*Barnesiellaceae*		−91.79	−203.62	20.03	0.11	−115.65	−234.15	2.88	0.06
*Gemellaceae*		−224.97	−420.95	−28.98	0.03	−238.58	−442.16	−35.00	0.023
*Micrococcaceae*		−259.40	−475.02	−43.76	0.02	−277.84	−498.70	−56.97	0.012

+ The model was adjusted for demographic measures, age, sex and BMI. A value of *p* < 0.05 was considered significant.

**Table 5 nutrients-14-00064-t005:** Multiple linear regression showing the combined contribution of families with cognition.

Bacterial Family	Cognition	F-Statistic	R^2^	Adjusted R^2^	*p* Value
*Carnobacteriaceae*	QESM	7.966	0.108	0.094	0.006
*Alcaligenaceae + Clostridiaceae*	QWM	6.973	0.177	0.151	0.002
*Bacteroidaceae + Barnesiellaceae + Rikenellaceae + Gemellaceae*	PoC	3.031	0.161	0.108	0.024
*Rikenellaceae + Clostridiaceae + Verrucomicrobiaceae*	CoA	2.039	0.088	0.045	0.118
*Bacteroidaceae + Barnesiellaceae + Gemellaceae + Micrococcaceae*	SoM	2.475	0.138	0.082	0.053

A multiple linear regression model was employed to combine the contributions of significantly correlated bacterial families to cognitive factors. A value of *p* < 0.05 was considered significant.

**Table 6 nutrients-14-00064-t006:** Spearman correlation between gut microbial function and cognition.

GBM	Cognition	*Rho*	*p* Value
Propionate Production III	CoA	−0.311	0.011
Tyrosine Degradation I	PoC	0.274	0.024
Phenylalanine Degradation		0.274	0.024
Tyrosine Degradation I	QWM	−0.246	0.045
Phenylalanine Degradation		−0.246	0.045

GBM, gut–brain module developed using omixer-rpmR. A value of *p* < 0.05 was considered significant.

## Data Availability

Data is contained within the article or Appendix A.

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
