# Peer review of "The Relationship between Gut Microbiome and Cognition in Older Australians"

_nutrients, 2021, doi:10.3390/nu14010064_

Round 1

Reviewer 1 Report

The authors present gut microbiome analysis of a small cohort of 63 older Australians showing the association of it with cognition. 

  1. It is well known that the microbiome is significantly affected by multiple factors, including, age, gender, nutritional status, family status, economic status, genetic factors etc. However, the authors did not consider these factors, and the sample size was small, which reduce the reliability of current results. 
  2. In table 1, how were the SD values calculated?
  3. Figure legend should support your figure entirely, meaning that the reader of your paper should be able to understand your figure, paired with its legend, without going to the results or methods sections to see what you say about your observations or how the experiment was done. 
  4. What statistical corrections were made for multiple comparisons (eg Bonferroni)?

Reviewer 2 Report

The paper is well written and topic interesting. The data are many and very suggestive. The "reading key" identified by the authors is correct and opens up  many future scenarios

Minor remarks

- title of paragraph 3.4: delete "?" after demographic

Reviewer 3 Report

The manuscript by Komanduri et al aims to examine the relationship between gut microbiome and cognition in older healthy people. The study is well executed and the results are clearly presented. However, in my opinion, introduction, as well as discussion, need to be modified. Introduction is very chaotic, needs to be specifically focused on the role of the microbiome on the gut-brain axis with special emphasis on aging. The discussion is limited to several statements and should be expanded. 

Round 2

Reviewer 1 Report

The authors have addressed my comments 

Reviewer 3 Report

the authors improved their manuscript